# Calculating the Effect of AlGaN Dielectric Layers in a Polarization Tunnel Junction on the Performance of AlGaN-Based Deep-Ultraviolet Light-Emitting Diodes

**DOI:** 10.3390/nano11123328

**Published:** 2021-12-07

**Authors:** Yong Wang, Zihui Zhang, Long Guo, Yuxuan Chen, Yahui Li, Zhanbin Qi, Jianwei Ben, Xiaojuan Sun, Dabing Li

**Affiliations:** 1State Key Laboratory of Luminescence and Applications, Changchun Institute of Optics, Fine Mechanics and Physics, Chinese Academy of Sciences, Changchun 130033, China; eeywang@ciomp.ac.cn (Y.W.); guolong17@mails.ucas.edu.cn (L.G.); chenyuxuan18@mails.ucas.ac.cn (Y.C.); 18730231183@163.com (Y.L.); qizhanbin2012@163.com (Z.Q.); benjianwei@ciomp.ac.cn (J.B.); sunxj@ciomp.ac.cn (X.S.); 2Center of Materials Science and Optoelectronics Engineering, University of Chinese Academy of Sciences, Beijing 100049, China; 3State Key Laboratory of High Power Semiconductor Lasers, Chang Chun University of Science and Technology, Changchun 130022, China; 4Key Laboratory of Electronic Materials and Devices of Tianjin, School of Electronics and Information Engineering, Hebei University of Technology, Tianjin 300401, China; zh.zhang@hebut.edu.cn; 5Key Laboratory of Reliability and Intelligence of Electrical Equipment, Hebei University of Technology, Tianjin 300401, China

**Keywords:** light emitting diodes, tunnel junction, polarization, AlGaN

## Abstract

In this work, AlGaN-based deep-ultraviolet (DUV) light-emitting diodes (LEDs) with AlGaN as the dielectric layers in p^+^-Al_0.55_Ga_0.45_N/AlGaN/n^+^-Al_0.55_Ga_0.45_N polarization tunnel junctions (PTJs) were modeled to promote carrier tunneling, suppress current crowding, avoid optical absorption, and further enhance the performance of LEDs. AlGaN with different Al contents in PTJs were optimized by APSYS software to investigate the effect of a polarization-induced electric field (*E_p_*) on hole tunneling in the PTJ. The results indicated that Al_0.7_Ga_0.3_N as a dielectric layer can realize a higher hole concentration and a higher radiative recombination rate in Multiple Quantum Wells (MQWs) than Al_0.4_Ga_0.6_N as the dielectric layer. In addition, Al_0.7_Ga_0.3_N as the dielectric layer has relatively high resistance, which can increase lateral current spreading and enhance the uniformity of the top emitting light of LEDs. However, the relatively high resistance of Al_0.7_Ga_0.3_N as the dielectric layer resulted in an increase in the forward voltage, so much higher biased voltage was required to enhance the hole tunneling efficiency of PTJ. Through the adoption of PTJs with Al_0.7_Ga_0.3_N as the dielectric layers, enhanced internal quantum efficiency (*IQE*) and optical output power will be possible.

## 1. Introduction

AlGaN-based deep-ultraviolet (DUV) light-emitting diodes (LEDs) have many advantages, including environmental protection, low power consumption, Hg-free material, compact size, controllable wavelength, and a long lifetime, and they can be applied in the fields of disinfection, sterilization, purification, biomedicine, gas sensing, optical data storage, non-line-of-sight communication, identification of hazardous biological agents and so on [1]. AlGaN-based DUV LEDs with a short wavelength of 210 nm [2] and an improved external quantum efficiency (EQE) of 20.3% (for 275 nm) [3] have been achieved. However, the EQE is still lower than that of GaN-based blue and green LEDs, and the EQE drops dramatically with decreasing wavelength [1]. As is well-known, the EQE is determined by the internal quantum efficiency (IQE) and the light extraction efficiency (LEE), where the IQE is determined by the material’s quality [4], the carrier injection [5], and the recombination [6]; and the LEE is determined by the total internal reflection (TIR) [7], the optical absorption [7], and the optical polarization [8,9].

In order to improve the EQE, high-efficiency carrier injection is required to realize high-performance AlGaN-based DUV LEDs. However, it is difficult to achieve high-efficiency doping, particularly p-type doping, in AlGaN with a high Al content [10,11]. In addition, AlGaN has much lower hole mobility than electron mobility (electron mobility of ~300 cm^2^/V·s and hole mobility of ~14 cm^2^/V·s at room temperature for unintentionally doped AlN [12,13]), which results in low hole injection efficiency and high electron leakage [5,14,15,16,17,18,19,20,21,22,23]. Moreover, poor ohmic contact between AlGaN and the electrode will increase the series resistance of an LED. Furthermore, the use of a horizontal electrode structure results in a very strong current crowding effect in the p-type side of an LED [24]. The vertical electrode structure is an alternative solution to suppress the current crowding effect [25,26]. However, there will be more difficulties and challenges in fabricating the vertical LEDs. In order to improve the current spreading and suppress the current crowding effect, some approaches, such as improving the design of the hole injection layer [27], optimization of the mesa area size to adjust the current distribution [28], enhancing the conductivity of the contact layer with the p-electrode [27], transparent p-side electrode fabrication [7,29], and use of a tunnel junction (TJ) or a polarization TJ (PTJ) [30,31,32,33,34,35,36,37,38,39,40,41] have been proposed.

A TJ is a heavily doped p^+^–n^+^ junction with a thickness of only some 10 nm. When an LED is operated at a forward biased voltage, the top TJ is operated at a reverse biased voltage, so the electrons can tunnel from the p^+^-side valence band into the n^+^-side conduction band. In order to reduce the resistance of the TJ, the heavy doping electron and hole concentrations are commonly over 1 × 10^20^ cm^−3^. A PTJ is a p^+^–i–n^+^ heterojunction, in which the built-in electric field (*E_b_*) is modulated by polarization induced electric field (*E_p_*) due to the existence of spontaneous polarization and piezoelectric polarization in the p^+^–i–n^+^ heterojunction. Therefore, the coupled electric field (*E_c_*) comes from the contributions of not only *E_b_* but also *E_p_*. The integration of a TJ or a PTJ in an LED can effectively enhance the hole injection efficiency, and PTJs can further promote hole tunneling by introducing *E_p_*. In addition, it can suppress the current crowding effect, improve lateral current spreading, and further enhance the uniformity of the top emitting light of the LED, which is possible through the contribution of its relatively high resistance. Moreover, it uses a low-resistivity n-type material instead of a high-resistivity p-type material as the top contact with the anode of the LED, which is very advantageous to ohmic contact. Moveover, it can be applied in multi-color LEDs [42,43], micro LEDs [44] and stacked laser diodes (LDs) [45]. Although there are many advantages, fabricating TJs or PTJs with heavy doping electron and hole concentrations (over 1 × 10^20^ cm^−3^) is very challenging.

In 2001, an InGaN-based blue LED with a p^+^-GaN/n^+^-GaN TJ was reported, with an optical output power about twice that of conventional LEDs without a TJ [32]. However, the forward voltage of the LED was accordingly increased from 3.9 to 4.9 V, which was caused by the relatively high resistance in TJ. A TJ has a positive impact on the lateral current spreading but has a negative impact on the forward voltage. Therefore, it is crucial to improve the tunneling efficiency by adopting PTJs. In 2013, an InGaN-based blue LED with a p^+^-GaN/In_0.15_Ga_0.85_N/n^+^-GaN PTJ was reported, and improved performance was achieved [35]. Polarization can induce high densities of positive and negative sheet charges at the p^+^-GaN/InGaN and InGaN/n^+^-GaN interfaces, which originate from spontaneous polarization and piezoelectric polarization in InGaN along the c-orientation [46]. The high density of polarization induced sheet charges at the p^+^-GaN/InGaN and InGaN/n^+^-GaN interfaces can build up *E_p_*. *E_p_* has a direction along the [000-1] orientation, in the same direction as *E_b_*. Therefore, *E_c_* comes from the contributions of not only *E_b_* but also *E_p_*, which can be enhanced by optimizing *E_p_* in the PTJ to improve the tunneling efficiency. The intensity of *E_c_* in a PTJ can be formulated as follows:(1)$$E=e\times \left|\frac{N_{dopant}\times L_{depletion}\pm \sigma_{p}}{\varepsilon_{r}\times \varepsilon_{0}}\right|$$
where *E* is the intensity of *E_c_* in PTJ, *e* is the value of a unit of electronic charge, *ε*_0_ is the absolute dielectric constant, *ε_r_* is the average relative dielectric constant for the PTJ, *N_dopant_* is the ionized dopant concentration in the space charge region, *L_depletion_* is the depletion region’s width, and *σ_p_* is the polarization-induced sheet charge density. The symbol “±” represents the direction of *E_p_*. The symbol “+” represents the same direction as *E_b_*, and the symbol “–” represents the reverse direction to *E_b_*. According to the expression in Equation (1), *E* can be determined by the factors *N_dopant_*, *L_depletion_*, *σ_p_*, and *ε_r_*.

It is necessary to point out that *ε_r_* is the inherent constant of a material. For InGaN, there is a linear relationship between *ε_r_* and the In content of InGaN, and *ε_r_* will become larger with an increase in the In content of InGaN. In addition, there is strong optical absorption in the PTJ when the In content of InGaN in the PTJ is higher than that in the active region, as it is quite disadvantageous to the LEE of an LED. AlGaN has a lower *ε_r_* than InGaN, and the *ε_r_* becomes small with an increase in the Al content of AlGaN. This means that AlGaN as the dielectric layer of a PTJ is superior to InGaN as the dielectric layer of a PTJ for enhancing the intensity of *E_c_*, promoting hole tunneling and avoiding optical absorption, as is very advantageous to the IQE and LEE of an AlGaN-based DUV LED.

However, there is still an inadequate view that AlGaN as the dielectric layer of PTJ cannot enhance the intensity of *E_c_*, due to the reverse direction of *E_p_* compared with *E_b_*. As a result, there are few reports on AlGaN as the dielectric layer of PTJs in LEDs. In 2017, InGaN-based near-ultraviolet (NUV) LEDs with p^+^-GaN/AlGaN/n^+^-GaN PTJs were optimized, and enhanced LED performance was achieved. This was the first report on the use of AlGaN as a dielectric material of a PTJ [36,37].

Additionally, the difficulty of heavy doping (a carrier concentration of over 1 × 10^20^ cm^−3^) in a PTJ has slowed down the progress of research into AlGaN-based DUV LEDs with PTJ. However, the research into AlGaN-based DUV LEDs with PTJs never stopped. In 2018, an AlGaN-based DUV LED with a p^+^-Al_0.65_Ga_0.35_N/In_0.2_Ga_0.8_N/graded n^+^-AlGaN PTJ (p^+^ = 5 × 10^19^ cm^−3^ and n^+^ = 1 × 10^20^ cm^−3^) was grown and fabricated, with a wavelength of 287 nm, a forward voltage of 10.5 V, and an optical output power of 54.4 W/cm^2^ at a current density of 1 kA/cm^2^ [34,35,36]. In 2021, an AlGaN-based DUV LED with a p^+^-GaN/n^+^-AlGaN PTJ was grown and fabricated, with a wavelength of 245 nm, a maximum EQE of 0.35% and a wall-plug efficiencies (WPE) of 0.21% [36]. It was proven that high-performance AlGaN-based DUV LEDs with PTJs can be achieved. However, InGaN or GaN as the dielectric layer of a PTJ is quite unsuitable for AlGaN-based DUV LED due to strong optical absorption in PTJ. AlGaN has a lower *ε_r_* and a lower optical absorption than InGaN. This means that AlGaN is superior to InGaN as the dielectric layer of PTJ for promoting hole tunneling, suppressing current crowding, increasing current spreading, and avoiding optical absorption of the PTJ in AlGaN-based DUV LEDs.

In this work, AlGaN-based DUV LEDs with AlGaN as the dielectric layers in p^+^-Al_0.55_Ga_0.45_N/AlGaN/n^+^-Al_0.55_Ga_0.45_N PTJs were proposed. AlGaN with different Al content in PTJs was optimized to investigate the effect of p^+^-Al_0.55_Ga_0.45_N/Al_x_GaN/n^+^-Al_0.55_Ga_0.45_N PTJs on the performance of AlGaN-based DUV LEDs, where x = 0.4, 0.55, and 0.7. The output characteristics of the AlGaN-based DUV LEDs including the forward voltage, optical output power, IQE and WPE were characterized. APSYS software was used to conduct a finite element analysis of the electrical, optical, and thermal properties of the LEDs.

## 2. Research Methods and Physical Models

AlGaN-based DUV LEDs for Structures A, B, and C were designed, as shown in Figure 1. The basic structures include a c-plane sapphire substrate, a 1 μm AlN buffer layer, a 2 μm n-Al_0.55_Ga_0.45_N layer (n = 5 × 10^19^ cm^−3^), a 50 nm n-AlGaN grading layer with the Al content ranging from 0.55 to 0.85 (n = 5 × 10^19^ cm^−3^), a 10 nm n-Al_0.85_Ga_0.15_N hole blocking layer (HBL) (n = 5 × 10^19^ cm^−3^), five pairs of multiple quantum wells (MQWs) with a 5 nm undoped Al_0.55_Ga_0.45_N quantum barrier (QB) and a 1.5 nm undoped Al_0.4_Ga_0.6_N quantum well (QW), a 10 nm p-Al_0.85_Ga_0.15_N electron blocking layer (EBL) (p = 5 × 10^19^ cm^−3^), a 50 nm p-AlGaN grading layer with the Al content ranging from 0.85 to 0.55 (p = 5 × 10^19^ cm^−3^), and a total of 322 nm for the top layers. For all structures, the bottom layers were the same and the 322 nm top layers were different. For Structure A (conventional TJ, no polarization layer), the top layers were an 11 nm p^+^-Al_0.55_Ga_0.45_N (p^+^ = 1 × 10^20^ cm^−3^)/11 nm n^+^-Al_0.55_Ga_0.45_N TJ (n^+^ = 1 × 10^20^ cm^−3^) and a 300 nm n-Al_0.55_Ga_0.45_N contact layer (n = 5 × 10^19^ cm^−3^). For Structure B (PTJ), the top layers were a 10 nm p^+^-Al_0.55_Ga_0.45_N (p^+^ = 1 × 10^20^ cm^−3^)/2 nm Al_0.4_Ga_0.6_N/10 nm n^+^-Al_0.55_Ga_0.45_N (n^+^ = 1 × 10^20^ cm^−3^) PTJ and a 300 nm n-Al_0.55_Ga_0.45_N contact layer (n = 5 × 10^19^ cm^−3^). For Structure C (PTJ), the top layers were a 10 nm p^+^-Al_0.55_Ga_0.45_N (p^+^ = 1 × 10^20^ cm^−3^)/2nm Al_0.7_Ga_0.3_N/10 nm n^+^-Al_0.55_Ga_0.45_N (n^+^ = 1 × 10^20^ cm^−3^) PTJ and a 300 nm n-Al_0.55_Ga_0.45_N contact layer (n = 5 × 10^19^ cm^−3^). Si and Mg were used to supply the electrons and holes in the n- and p-layers and the TJ. The anode of the LED was the ohmic contact with the top layer, and the mesa size was set to 300 × 300 μm^2^, with an anode size of 300 × 50 μm^2^ and a cathode size of 300 × 70 μm^2^.

APSYS software (APSYS-2017, Crosslight Software Inc. Vancouver, Canada) was used to simulate the LEDs. In the simulation, the current continuity equations, Poisson equations, and Schrödinger equations would be solved with proper boundary conditions. The following parameters were set: an Auger recombination coefficient of 1 × 10^−42^ m^6^/s [35,47], a Shockley–Read–Hall (SRH) lifetime of 1 × 10^−8^ s [48,49], an energy band offset ratio (ΔE_C_/ΔE_V_) of 0.5 [50,51], a [0001] polarization level of 40% [46,52,53], effective masses of the electrons and holes for GaN and AlN [54], and an ambient temperature of 300 K. The effective masses of the tunneling particles in the TJs were 0.230 *m*_0_, 0.225 *m*_0_, and 0.234 *m*_0_ for Structures A, B, and C, respectively; *m*_0_ is the free electron mass. Other parameters of III-nitride based semiconductors can be found elsewhere [55].

## 3. Results and Analysis

Figure 2 shows the distributions of the electric fields (Figure 2a–c), the charged state densities of the space charges (Figure 2d), and the charged state densities of the fixed space charges at the interfaces (Figure 2e) in the TJs for Structures A, B, and C at a bias voltage of 0 V. *N* is the charged state density of the space charge.

For Structure A, the TJ is a p^+^–n^+^ homojunction and *E_c_* comes from the contribution of only *E_b_*, as shown in Figure 2a. *E_b_* comes from the ionized dopant in the space charge region; has a direction along the [000-1] orientation, beginning with *N*_11_ in the n^+^-Al_0.55_Ga_0.45_N layer and ending with *N*_12_ in the p^+^-Al_0.55_Ga_0.45_N layer; and can be calculated as *N*_11_ = 1.0 × 10^20^ cm^−3^ and *N*_12_ = −1.0 × 10^20^ cm^−3^, as shown in Figure 2d.

For Structure B, the PTJ is a p^+^–i–n^+^ heterojunction, and *E_c_* comes from the contributions of *E_b_* and *E_p_*, as shown in Figure 2b. *E_b_* has a direction along the [000-1] orientation, beginning with *N*_21_ in the n^+^-Al_0.55_Ga_0.45_N layer and ending with *N*_22_ in the p^+^-Al_0.55_Ga_0.45_N layer, and it can be calculated as *N*_21_ = 1.0 × 10^20^ cm^−3^ and *N*_22_ = −1.0 × 10^20^ cm^−3^, as shown in Figure 2d. *E_p_* is built up from the differences between *N*_3_ at the Al_0.4_Ga_0.6_N/n^+^-Al_0.55_Ga_0.45_N interface and *N*_4_ at the p^+^-Al_0.55_Ga_0.45_N/Al_0.4_Ga_0.6_N interface, where *N*_3_ > 0 and *N*_4_ < 0 under compressive strain of Al_0.4_Ga_0.6_N, and can be calculated as *N*_3_ = 2.9 × 10^20^ cm^−3^ and *N*_4_ = −3.1 × 10^20^ cm^−3^, as shown in Figure 2e. *E_p_* has a direction along the [000-1] orientation in the Al_0.4_Ga_0.6_N layer, in the same direction as *E_b_*, and a direction along the [0001] orientation at both sides of the Al_0.4_Ga_0.6_N layer, in the reverse direction to *E_b_*, as shown in Figure 2b.

For Structure C, the PTJ is also a p^+^–i–n^+^ heterojunction, and *E_c_* also comes from the contributions of *E_b_* and *E_p_*, as shown in Figure 2c. *E_b_* has a direction along the [000-1] orientation, beginning with *N*_31_ in the n^+^-Al_0.55_Ga_0.45_N layer and ending with *N*_32_ in the p^+^-Al_0.55_Ga_0.45_N layer, and can be calculated as *N*_31_ = 1.0 × 10^20^ cm^−3^ and *N*_32_ = −1.0 × 10^20^ cm^−3^ as shown in Figure 2d. *E_p_* is built up from the differences between *N*_5_ at the p^+^-Al_0.55_Ga_0.45_N/Al_0.7_Ga_0.3_N interface and *N*_6_ at the Al_0.7_Ga_0.3_N/n^+^-Al_0.55_Ga_0.45_N interface, where *N*_5_ > 0 and *N*_6_ < 0 under tensile strain of Al_0.7_Ga_0.3_N, and can be calculated as *N*_5_ = 3.7 × 10^20^ cm^−3^ and *N*_6_ = −3.4 × 10^20^ cm^−3^, as shown in Figure 2e. *E_p_* has a direction along the [0001] orientation in the Al_0.7_Ga_0.3_N layer, in the reverse direction to *E_b_*, and a direction along the [000-1] orientation at both sides of the Al_0.7_Ga_0.3_N layer, in the same direction as *E_b_*, as shown in Figure 2c.

Figure 3 compares the electric field profiles at a relative horizontal position of 100 μm in the TJs of LEDs (300 × 300 μm^2^) for Structures A, B, and C at a current of 180 mA. For all three structures, the intensities of *E_c_* were very strong in the TJ regions. For Structure A, *E_c_* comes from the contribution of only *E_b_* with a direction along the [000-1] orientation, as shown in Figure 2a. For Structure B, *E_c_* comes from the contributions of *E_b_* and *E_p_*. *E_b_* has a direction along the [000-1] orientation. *E_p_* has a direction along the [000-1] orientation in the Al_0.4_Ga_0.6_N layer, in the same direction as *E_b_*, and a direction along the [0001] orientation in the p^+^-Al_0.55_Ga_0.45_N and n^+^-Al_0.55_Ga_0.45_N layers, in the reverse direction to *E_b_*, as shown in Figure 2b. As a result, the intensity of *E_c_* is enhanced by *E_p_* in the Al_0.4_Ga_0.6_N layer, and degraded by *E_p_* in the p^+^-Al_0.55_Ga_0.45_N and n^+^-Al_0.55_Ga_0.45_N layers compared with Structure A, as shown in Figure 3. For Structure C, *E_c_* comes from the contributions of *E_b_* and *E_p_*. *E_b_* has a direction along the [000-1] orientation. *E_p_* has a direction along the [0001] orientation in the Al_0.7_Ga_0.3_N layer, in the reverse direction to *E_b_*, and a direction along the [000-1] orientation in the p^+^-Al_0.55_Ga_0.45_N and n^+^-Al_0.55_Ga_0.45_N layers, in the same direction as *E_b_*, as shown in Figure 2c. As a result, the intensity of *E_c_* is degraded by *E_p_* in the Al_0.7_Ga_0.3_N layer, and enhanced by *E_p_* in the p^+^-Al_0.55_Ga_0.45_N and n^+^-Al_0.55_Ga_0.45_N layers, as compared with Structure A, as shown in Figure 3.

Moreover, *ε_r_* has a role in controlling the intensity of *E_c_*, according to the expression in Equation (1). For AlGaN, *ε_r_* becomes small with an increase in the Al content of AlGaN. For Structure B, Al_0.4_Ga_0.6_N as the dielectric layer has a higher *ε_r_* than Structure A with Al_0.55_Ga_0.45_N, which should have decreased the intensity of *E_c_*. Nevertheless, an increased electric field is still obtained because of the contribution of a polarization-induced electric field in the center. For Structure C, Al_0.7_Ga_0.3_N as the dielectric layer has a lower *ε_r_* than Structure A with Al_0.55_Ga_0.45_N, which enhances the intensity of *E_c_* because of the contribution of *ε_r_* in the center. Thus, according to the final results regarding the contributions of *E_b_*, *E_p_*, and *ε_r_*, the intensities of *E_c_* are in the following order: Structure B > Structure C > Structure A in the center peak, and Structure C > Structure A > Structure B at both sides of the center, as shown in Figure 3.

According to Figure 3, the calculated peak intensities of *E_c_* are as follows: *E_A_* = 7.40 × 10^6^ V/cm, *E_B_* = 9.07 × 10^6^ V/cm, and *E_C_* = 8.24 × 10^6^ V/cm for Structures A, B, and C in the center peak, respectively. This means that very strong electric field intensities are generated in the TJ regions of the LEDs in Structures A, B, and C. Driven by *E_c_*, the electrons in the valence band of the p^+^-Al_0.55_Ga_0.45_N layer can tunnel through the TJ and inject themselves into the conduction band of the n^+^-Al_0.55_Ga_0.45_N layer. The holes are simultaneously generated in the valence band of the p^+^-Al_0.55_Ga_0.45_N layer, and then injected into the MQWs of the LEDs.

Figure 4 shows a comparison of the energy-band profiles at a relative horizontal position of 100 μm in the TJs of LEDs (300 × 300 μm^2^) for Structures A, B, and C, at a current of 180 mA. For Structures A, B, and C, the conduction bands of n^+^-Al_0.55_Ga_0.45_N (n^+^ = 1 × 10^20^ cm^−3^) align well below the valence bands of p^+^-Al_0.55_Ga_0.45_N (p^+^ = 1 × 10^20^ cm^−3^). When the LEDs are operated at a forward biased voltage, the top TJs are operated at a reverse biased voltage, so the electrons can tunnel from the p^+^-side valence band into the n^+^-side conduction band, and the holes are simultaneously generated in the p-type layer. Under the reverse biased voltage, the conduction band levels in the TJs are totally different. The n^+^-side conduction band levels are in the following order: Structure B > Structure A > Structure C. The widths of the TJs in Structures A, B, and C are the distances between Positions O and A, Positions O and B, and Positions O and C, respectively. The calculated widths of the TJs are as follows: |OA| = 8.9 nm, |OB| = 9.5 nm, and |OC| = 8.0 nm. The widths of the TJs for Structures A, B, and C are in the following order: Structure B > Structure A > Structure C. The short width of the PTJ in Structure C can counteract the large energy barrier, and may help to enhance the tunneling probabilities of the electrons in the PTJ.

Figure 5a shows a comparison of the hole concentration distributions along the vertical direction at a relative horizontal position of 100 μm in the MQWs of the LEDs for Structures A, B, and C, at a current of 180 mA. The hole concentrations along the vertical direction in the MQWs are in the following order: Structure C > Structure A > Structure B. The high hole concentration in the MQWs in Structure C comes from the enhanced lateral current spreading in the PTJ.

Figure 5b shows a comparison of the radiative recombination rates along the vertical direction at a relative horizontal position of 100 μm in the MQWs of the LEDs for Structures A, B, and C at a current of 180 mA. The radiative recombination rates along the vertical direction in the MQWs are in the following order: Structure C > Structure A > Structure B. The radiative recombination rates come from the contributions of the high electron and hole concentrations in the MQWs, as is consistent with the hole-concentration distributions in the MQWs, as shown in Figure 5a.

In order to investigate the effects of the TJs on the current spreading in LEDs, the lateral distributions of the hole concentrations and the radiative recombination rates along the horizontal direction (x-axis) in the LEDs were characterized. Figure 5c shows a comparison of the lateral distributions of the hole concentrations and the radiative recombination rates along the horizontal direction (x-axis) and at a relative vertical position of the fifth QW (c-axis) in LEDs for Structures A, B, and C. For Structure B, the lateral distributions of the hole concentrations and the radiative recombination rates along the horizontal direction in the LEDs are very non-uniform: high at the horizontal positions of 0~50 μm, which is just below the anode, and low at the positions of 50~200 μm. This means that the positions under the anode are the main current flow paths going through the MQWs, which will cause current crowding and Joule heat. For Structures A, B, and C, the lateral distribution uniformities of the hole concentrations and the radiative recombination rates along the horizontal direction (x-axis) in the LEDs are in the following order: Structure C > Structure A > Structure B. The uniformities come from the contributions of high resistance in the TJs. If we compare Structures A, B, and C, the tunneling probability of electrons and the Al content of AlGaN in the TJs determine the resistance. Structure C has a lower electron tunneling probability and a higher Al content of AlGaN in the PTJ than Structure B, thus resulting in the higher resistance of the PTJ in Structure C. Therefore, Structure C with Al_0.7_Ga_0.3_N as the dielectric layer has higher uniformity than Structure B with Al_0.4_Ga_0.6_N as the dielectric layer, which means that relatively higher resistance of the TJ has a meaningful impact on the current spreading of an LED.

In order to quantitatively compare the lateral distribution uniformities of the hole concentrations and radiative recombination rates on the output characteristics of the LEDs for Structures A, B, and C, the integrated intensities of the lateral distributions of the hole concentrations and radiative recombination rates at the relative horizontal positions between 0 and 200 μm were characterized. Figure 5d shows a comparison of the integrated intensities of the lateral distributions of the hole concentrations and radiative recombination rates along the horizontal direction (x-axis) and at a relative vertical position of the fifth QW (c-axis) in LEDs for Structures A, B, and C. The integrated intensities for both the hole concentrations and radiative recombination rates are in the following order: Structure C > Structure A > Structure B. Compared with the lateral distributions of the hole concentrations and radiative recombination rates along the horizontal direction (x-axis), the integrated intensities more accurately reflected the output characteristics of LEDs with the different structures.

Figure 6 shows a comparison of the output current–voltage (*I*–*V*) characteristics of the LEDs (300 × 300 μm^2^) for Structures A, B, and C. The forward voltages of the LEDs are in the following order: Structure C > Structure A > Structure B. When an LED is operated at a forward biased voltage, the top TJ is operated at a reverse biased voltage. A TJ operating at a reverse biased voltage can be treated as a resistor, and its resistance determines the forward voltage of the LED, so that the forward voltage of an LED increases with an increase in the resistance of the JT. If we compare Structures A, B, and C, the tunneling probability of electrons and the Al content of AlGaN in the TJs determine the resistance, and the resistance determines the forward voltage of the LED. The resistances of the TJs are in the following order: Structure C > Structure A > Structure B, so the forward voltages of the LEDs are in the following order: Structure C > Structure A > Structure B. This indicates that the TJ has a nonnegligible impact on the forward voltage of DUV LEDs, and much higher biased voltage is required to realize higher tunneling efficiency for Structure C with a higher Al content of AlGaN.

Figure 7a shows a comparison of the IQE and optical output power of the LEDs (300 × 300 μm^2^) for Structures A, B, and C. Both the IQE and the optical output power of the LEDs are in the following order: Structure C > Structure A > Structure B, consistent with the results shown in Figure 5d, and inditcating that the enhanced IQE and optical output power of the LED for Structure C were achieved. Figure 7b shows a comparison of the WPE of LEDs for Structures A, B, and C. The WPE of the LEDs is in the following order: Structure B > Structure A > Structure C. The WPE is inversely proportional to the forward voltage, so the LED for Structure C had the lowest WPE due to the highest forward voltage.

## 4. Conclusions

AlGaN-based DUV LEDs with AlGaN as the dielectric layers in p^+^-Al_0.55_Ga_0.45_N/AlGaN/n^+^-Al_0.55_Ga_0.45_N PTJs were proposed for promoting carrier tunneling, suppressing current crowding, avoiding optical absorption, and further enhancing the performance of LEDs. Al_0.7_Ga_0.3_N with a lower dielectric constant can realize a higher hole concentration and a higher radiative recombination rate in MQWs than Al_0.4_Ga_0.6_N with a higher dielectric constant. The high hole concentration and high radiative recombination rate in MQWs come from enhanced lateral current spreading in the LED. Moreover, besides the high electrically conductive n^+^-Al_0.55_Ga_0.45_N layer in the PTJ, Al_0.7_Ga_0.3_N as the dielectric layer has relatively high resistance, which can increase the lateral current spreading and enhance the uniformity of the top emitting light of AlGaN-based DUV LEDs. As a result, the IQE and optical output power can be enhanced. However, its relatively high resistance results in an increase in the forward voltage, which is disadvantageous to the WPE. Through the adoption of PTJ with Al_0.7_Ga_0.3_N as the dielectric layers, enhanced IQE and optical output power can be achieved. It is strongly believed that the proposed structure is promising for further improving the IQEs of DUV LEDs, and the physics of the reported device is also useful for better understanding the carrier tunnel, transport and recombination mechanisms of DUV LEDs.

## Figures and Tables

**Figure 1 nanomaterials-11-03328-f001:**
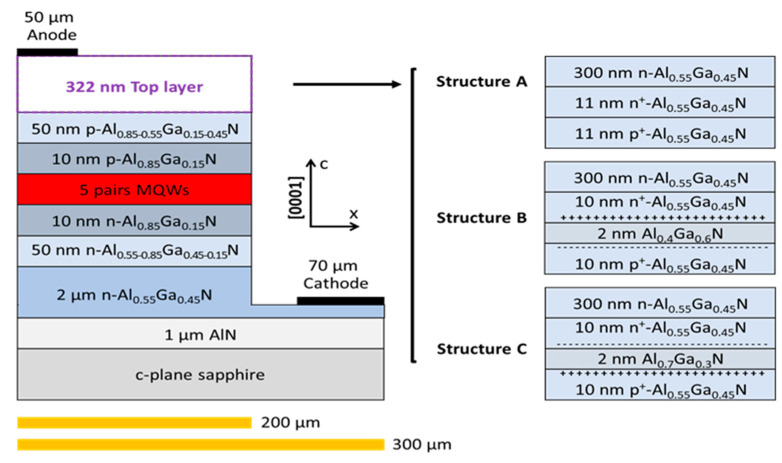
LED structures and the different top layers of Structures A, B, and C.

**Figure 2 nanomaterials-11-03328-f002:**
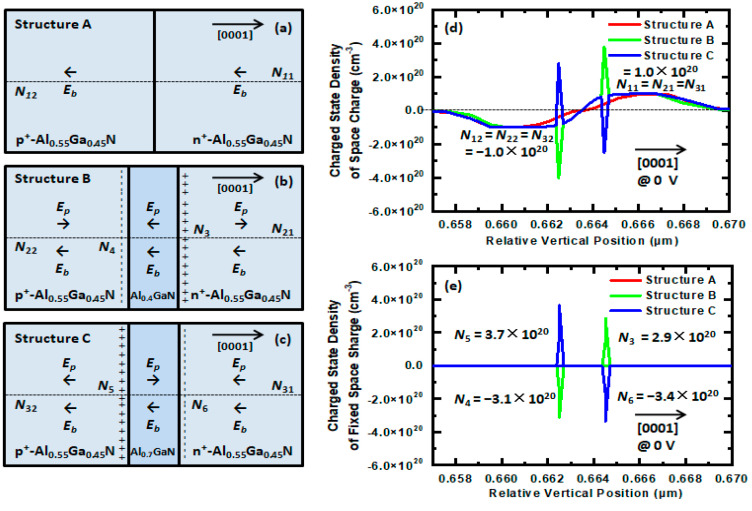
Distributions of the electric fields (**a**–**c**), the charged state densities of the space charges (**d**) and the charged state densities of the fixed space charges at the interfaces (**e**) in TJs in Structures A, B, and C at a bias voltage of 0 V. *N* is the charged state density of the space charge.

**Figure 3 nanomaterials-11-03328-f003:**
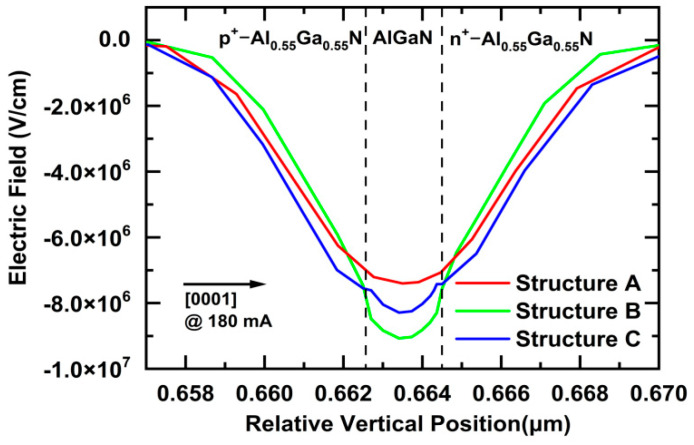
Comparison of the electric field profiles at a relative horizontal position of 100 μm in the TJs of LEDs (300 × 300 μm^2^) in Structures A, B, and C at a current of 180 mA.

**Figure 4 nanomaterials-11-03328-f004:**
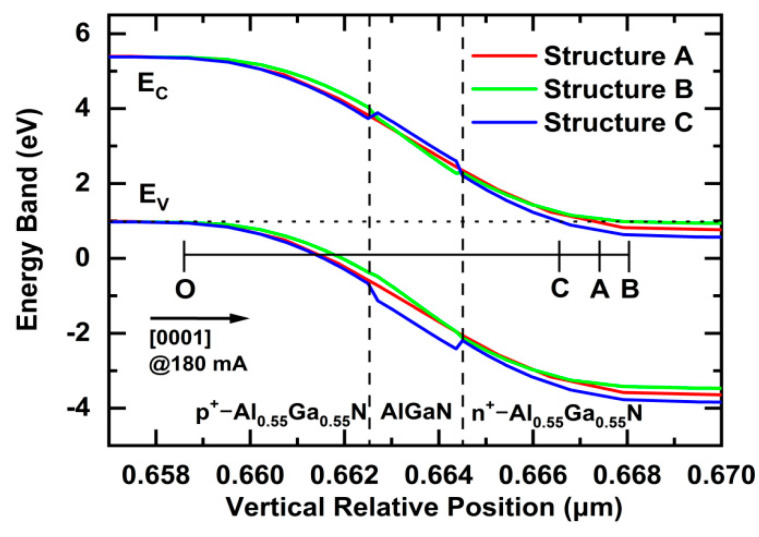
Comparisons of the energy band profiles at a relative horizontal position of 100 μm in the TJs of LEDs (300 × 300 μm^2^) in Structures A, B, and C at a current of 180 mA.

**Figure 5 nanomaterials-11-03328-f005:**
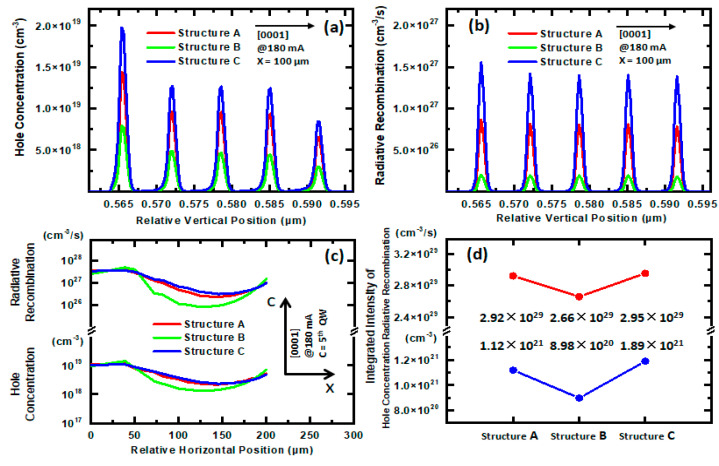
(**a**) Hole concentrations and (**b**) radiative recombination rates in the MQWs. (**c**) Lateral distributions of the hole concentrations and radiative recombination rates along the horizontal direction (x-axis) in MQWs. (**d**) Integrated values of lateral hole concentrations and lateral radiative recombination rates at the relative horizontal positions between 0 and 200 μm for Structures A, B, and C at a current of 180 mA.

**Figure 6 nanomaterials-11-03328-f006:**
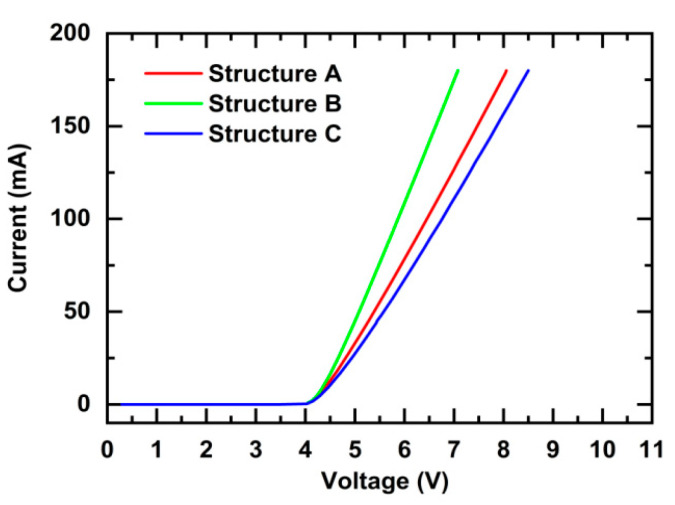
Comparison of the output *I*–*V* characteristics of LEDs (300 × 300 μm^2^) for Structures A, B, and C.

**Figure 7 nanomaterials-11-03328-f007:**
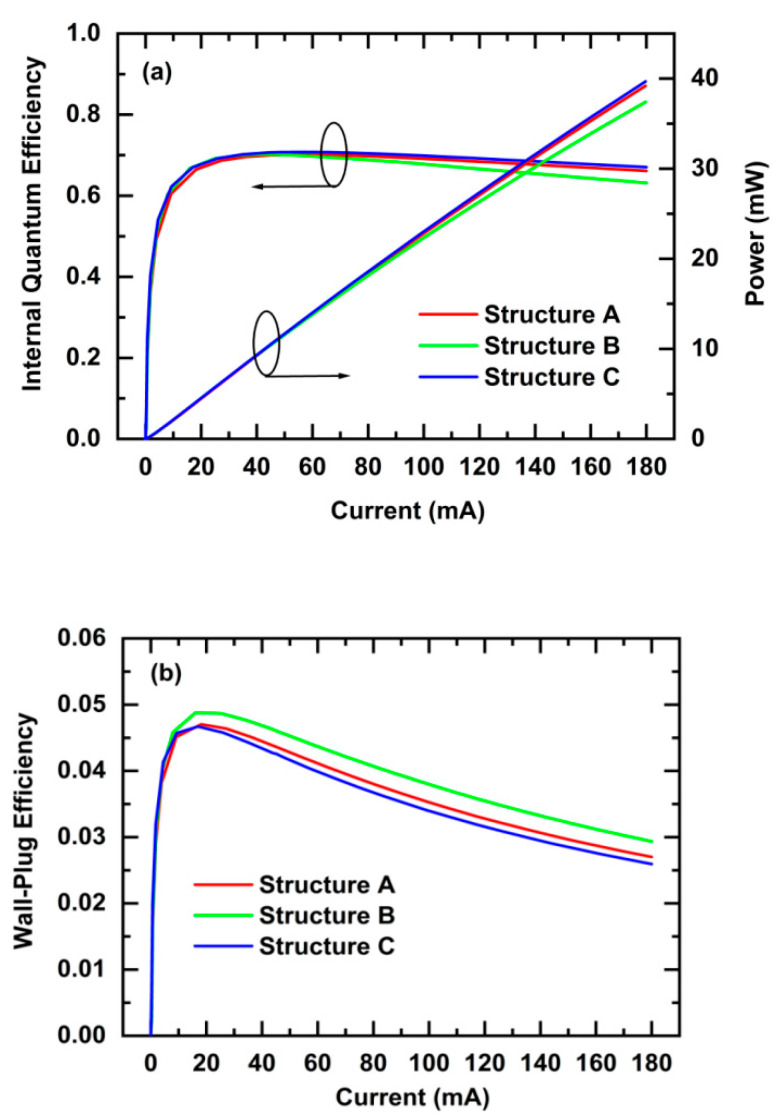
Comparison of the IQE and optical output power (**a**) and the WPE (**b**) of the LEDs (300 × 300 μm^2^) for Structures A, B, and C.

## Data Availability

The data presented in this study are available on request from the corresponding author.

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
