# Peer review of "Calculating the Effect of AlGaN Dielectric Layers in a Polarization Tunnel Junction on the Performance of AlGaN-Based Deep-Ultraviolet Light-Emitting Diodes"

_nanomaterials, 2021, doi:10.3390/nano11123328_

Round 1
Reviewer 1 Report
Dear Editor, dear Authors
This manuscript is devoted to modeling of AlGaN/AlN structures that can be UV light emitting diodes. The work is good and can be useful for growers that would like to try to make such LEDs. The main problem is that I was misled by title and abstract suggesting that authors actually made such diodes. I think that the paper can be published but wording of the title and abstract should change.
Instead of "Effect of AlGaN dielectric layer in polarization tunnel junction on performance..",
it could be: "Calculation of effect of AlGaN dielectric layer in polarization tunnel junction on performance.."
In abstract, line 21: "are investigated" change to "modeled".
In abstract, line 32: "achieved" change to "predicted".
There is huge difference between calculation and real device. I calculated similar things on a 1000$ computer but our MOCVD was about 1,000,000$.
Other comments:
- Page 2, line 47.
It is not true that "the LEE is determined by the optical polarization [7-8] and optical absorption [9]". Both these effects are minor. The LEE is determined mainly by high refractive index that causes reflection of light back to the device.
- Word "structure" should be written from small letter. And these are "the structures" not "Structures". (lines: 152, 172 and many other).
- Page 5, line 221. Should be Figure 2.
- Instead of "in order of" it is better to use "in the following order". (lines: 243, 256, 270, and other)
- Page 8, line 333, about TJ:
"treated as a resistor with linear I-V characteristic"
There is always some voltage drop on the tunnel junction, so it cannot be treated just like ohmic contact.
- Page 10, line358.
Instead of "PTJs are investigated to promote". It would be better: "PTJs are proposed to promote".
Author Response
Response Letter
The authors give special thanks to the referees for their meaningful guidance and constructive suggestions. The manuscript has been revised according to the referee’s comments, and the detailed revision is described in the response to referees. Based on the referees’ comments, the opinions are answered as follows.
(1) Response to Referee 1:
1) I think that the paper can be published but wording of the title and abstract should change. Instead of "Effect of AlGaN dielectric layer in polarization tunnel junction on performance..", it could be: "Calculation of effect of AlGaN dielectric layer in polarization tunnel junction on performance.."
Answer:
Thank you for your suggestion. In the revised manuscript, the title has been revised from “Effect of AlGaN dielectric layer in polarization tunnel junction on performance of AlGaN-based deep ultraviolet light emitting diode” to “Calculation of effect of AlGaN dielectric layer in polarization tunnel junction on performance of AlGaN-based deep ultraviolet light emitting diode”.
2) In abstract, line 21: "are investigated" change to "modeled".
Answer:
Thank you for your suggestion. In the revised manuscript, the word of “investigated” has been revised to “modelled”.
3) In abstract, line 32: "achieved" change to "predicted".
Answer:
Thank you for your suggestion. In the revised manuscript, the word of “achieved” has been revised to “predicted”.
4) In introduction, Page 2, line 47: It is not true that "the LEE is determined by the optical polarization [7-8] and optical absorption [9]". Both these effects are minor. The LEE is determined mainly by high refractive index that causes reflection of light back to the device.
Answer:
Thank you for your suggestion. In the revised manuscript, the expression of "the LEE is determined by the optical polarization [7-8] and optical absorption [9]" has been revised to “the LEE is determined by the total internal reflection (TIR) [7], optical absorption [7] and optical polarization [8-9].”
5) Word "structure" should be written from small letter. And these are "the structures" not "Structures". (lines: 152, 172 and many other).
Answer:
Thank you for your suggestion. In the revised manuscript, all these expressions about “Structure” and "Structures" have been revised to "structure" and "structures", respectively.
6) Page 5, line 221: Should be Figure 2.
Answer:
Thank you for your suggestion. In the revised manuscript, Figure 1 has been revised to Figure 2.
7) In lines 243, 256, 270, and other lines: Instead of "in order of" it is better to use "in the following order".
Answer:
Thank you for your suggestion. In the revised manuscript, all these expressions about “in order of” have been revised to "in the following order".
8) Page 8, line 333, about TJ: "treated as a resistor with linear I-V characteristic". There is always some voltage drop on the tunnel junction, so it cannot be treated just like ohmic contact.
Answer:
Thank you for your suggestion. In the revised manuscript, the expression of “The TJ operated at reverse biased voltage can be treated as a resistor with linear I-V characteristic” has been revised to “The TJ operated at reverse biased voltage can be treated as a resistor”, and the expression of “with linear I-V characteristic” is deleted. Therefore, the TJ is just a resistor, not a resistor like ohmic contact with linear I-V characteristic.
9) In Page 10, line358: Instead of "PTJs are investigated to promote". It would be better: "PTJs are proposed to promote".
Answer:
Thank you for your suggestion. In the revised manuscript, the expression of "PTJs are investigated to promote" has been revised to "PTJs are proposed to promote".

Reviewer 2 Report
This manuscript describes some theoretical studies about possible AlGaN tunnel junction layer structures for the p-contact of AlGaN-based UV LED structures. This is a very interesting subject, and significant progress on this field would be more than welcome.
However, I have strong doubts whether the methods and results described in this manuscript are really helpful. Moreover, I have detected some strong flaws in this manuscript. Let me be somewhat more specific:
1) For using the tunnel effect in such structures, a large tunneling probability must be realized. This tunneling probability depends mainly on the energetic height, the local width of the tunnel junction (better to say: The details of the shape of the tunneling barrier) and the effective mass of the tunneling particles. I cannot see that these parameters have been adequately taken into account in this manuscript.
2) The authors claim that the tunneling current is mainly governed by their equation 2 (work performed on a hole, depending only on initial energetic state and final state). This is certainly not correct, as it does not consider the shape of the tunnel barrier. Hence this equation is meaningless here!
3) Moreover, this equation is formally incorrect, as the integral does not result in an energy.
4) The authors have found best tunneling properties for sample C which contains a larger band gap material in the center, i.e. the tunneling barrier is higher and consequently the tunneling probability is certainly smaller than for structures A and B.
5) According to their calculations, the tunneling distance is in the range of 9nm (Fig. 4), a distance which is typically too large for efficient tunneling, particularly for heavy holes.
6) The authors must check carefully their manuscript about errors. In particular and as an example, announcing specific charge density of more than 1e20 C/cm3 (Coulomb/cm3) in the text on page 4, chapter 3, and in Fig. 1 is just nonsense.
7) The authors find a larger series resistance for the structure C which only differs by a thin layer of Al0.7Ga0.3N (thickness 2nm) in the center and argue that this increase is due to the low conductivity of this layer. Please check what would be the influence of such a thin layer on the series resistance if only the carrier mobility would be the reason (as the authors argue). It remains unclear why structure C shows the highest series resistance (which is certainly correct, but owing to other reasons than argued by the authors) but provides, as the authors say, best tunnel current probability. This is inconsistent!
8) It remains unclear why the authors have chosen a current of 180mA and what that really means. If any current value is of importance here, then it should be announced as current density, not as absolute current, as far as the device area is not fixed.
9) The manuscript is written in very bad English language. If reconsidered for any further submission, then it must be corrected preferably by a native scientific speaker!
10) I have annotated a lot of additional problems as marginal notes in the pdf manuscript for further consideration by the authors.
11) This manuscript describes pure theoretical work without any experimental confirmation. This cannot be easily seen by the reader as it is neither mentioned in the title nor in the abstract, but only at the very end of the introduction.
In conclusion, I cannot see how this manuscript can be easily improved concerning its scientific content. Therefore I propose to reject the manuscript.

Author Response
Response Letter
The authors give special thanks to the referees for their meaningful guidance and constructive suggestions. The manuscript has been revised according to the referee’s comments, and the detailed revision is described in the response to referees. Based on the referees’ comments, the opinions are answered as follows.
(2) Response to Referee 2:
1) For using the tunnel effect in such structures, a large tunneling probability must be realized. This tunneling probability depends mainly on the energetic height, the local width of the tunnel junction (better to say: The details of the shape of the tunneling barrier) and the effective mass of the tunneling particles. I cannot see that these parameters have been adequately taken into account in this manuscript.
Answer:
(I) The authors thank the referee for the comments, and we are sorry for the missed information. However, all these parameters have been taken into account. In our models, the barrier height of energy band and the width of tunnel width can be calculated by solving the Poisson equation. Here, it is worth showing the effective masses of tunneling particles, which are 0.230 m0, 0.225 m0, and 0.234 m0 for the TJs of structures A, B and C, respectively.
(II) Hence, the following description is added in the revised manuscript as follow.
“The effective masses of tunneling particles in TJs are 0.230 m0, 0.225 m0, and 0.234 m0 for strcutures A, B and C, respectively. Note, m0 is the free electron mass.”
2) The authors claim that the tunneling current is mainly governed by their equation 2 (work performed on a hole, depending only on initial energetic state and final state). This is certainly not correct, as it does not consider the shape of the tunnel barrier. Hence this equation is meaningless here!
Answer:
(I) The authors thank the referee for the comments. In the revised manuscript, the original expression (2) about the work done by Ec in PTJ on a hole has been deleted.
(II) Instead, the shape of the tunnel barrier is reflected by the electric field intensity in the TJ. The peak intensities of Ec in TJs with structures A, B and C, are calculated, and the process of electrons tunneling through a TJ under the drive of Ec are discussed.
The added paragraphs in manuscript are as follows.
“According to Figure 3, the calculated peak intensities of Ec are as follows: EA = 7.40 ´ 106 V/cm, EB = 9.07 ´ 106 V/cm, and EC = 8.24 ´ 106 V/cm, for structures A, B and C in the center peak, respectively. It means that the very strong electric field intensities are generated in TJs regions of LEDs with structures A, B and C. Under the drive of Ec, the electrons in valence band of p+-Al0.55Ga0.45N layer can tunnel through TJ, and inject into conduction band of n+-Al0.55Ga0.45N layer. The holes are simultaneously generated in valence band of p+-Al0.55Ga0.45N layer, and then injected into the MQWs of LEDs.”
3) Moreover, this equation is formally incorrect, as the integral does not result in an energy.
Answer:
The authors thank the referee for the comments. In the revised manuscript, the original expression (2) about the work done by the Ec in PTJ on a hole has been deleted.
4) The authors have found best tunneling properties for sample C which contains a larger band gap material in the center, i.e. the tunneling barrier is higher and consequently the tunneling probability is certainly smaller than for structures A and B.
Answer:
(I) The authors thank the referee for the comments. For structure C, although the peak intensity of Ec is not the strongest, however the carriers have the largest lateral spreading, which increases a large lateral distribution uniformity of hole concentration and radiative recombination rate, and results in the enhancement of IQE and optical output power of LED.
(II) For structure B, the peak intensity of Ec in the TJ region is the strongest, so that the carriers travel downwards by interband tunneling before they effectively spread to the mesa edges. Therefore, the current cannot be sufficiently spread, and the hole concentration in the region away from the p-electrode becomes low, which results in the decrease of IQE and optical output power of LED.
5) According to their calculations, the tunneling distance is in the range of 9nm (Fig. 4), a distance which is typically too large for efficient tunneling, particularly for heavy holes.
Answer:
The authors thank the referee for the comments. According to the calculation on tunneling distance in Figure 4, the calculated widths of TJs with structures A, B and C are as follows: |OA| = 8.9 nm, |OB| = 9.5 nm, and |OC| = 8.0 nm, respectively.
The values are true. The reasons are as follows.
(I) The values come from the data in Figure 4.
(II) The relatively larger width of TJ is caused by relatively wider bandgap of AlGaN.
(III) Under the drive of Ec, the electrons in the valence band of the p+-Al0.55Ga0.45N layer can tunnel through TJ before arriving at the conduction band of the n+-Al0.55Ga0.45N layer, and the holes are simultaneously generated in p+-Al0.55Ga0.45N layer. Therefore, the electrons participate the interband tunneling process. However, the effective mass for the tunneling particles is decided by the electron (me) mass and the heavy hole mass (mhh). The referee is correct that if the effective mass is merely determined by the heavy hole mass, then the interband tunneling process will not take place in the TJ with tunneling distance is in the range of 9 nm.
6) The authors must check carefully their manuscript about errors. In particular and as an example, announcing specific charge density of more than 1e20 C/cm3 (Coulomb/cm3) in the text on page 4, chapter 3, and in Fig. 1 is just nonsense.
Answer:
(I) The authors thank the referee for the comments, and we are sorry for these errors. In fact, there are the writing errors.
(II) In the revised manuscript, these values are revised from ρ1 = 1.0 ´ 1020 C/cm3 to ρ1 = 1.0 ´ 1020 cm-3. So are others.
7) The authors find a larger series resistance for the structure C which only differs by a thin layer of Al0.7Ga0.3N (thickness 2nm) in the center and argue that this increase is due to the low conductivity of this layer. Please check what would be the influence of such a thin layer on the series resistance if only the carrier mobility would be the reason (as the authors argue). It remains unclear why structure C shows the highest series resistance (which is certainly correct, but owing to other reasons than argued by the authors) but provides, as the authors say, best tunnel current probability. This is inconsistent!
Answer:
(I) The authors thank the referee for the comments. It is a fact that structure C shows the highest series resistance.
(II) Actually, the intensity of Ec in the TJ with structure B is the largest, which causes the smallest forward voltage according to Figure 6.
(III) Although the intensity of Ec in TJ with structure C is larger than that with structure A, the forward voltage of LED with structure C is not reduced according to Figure 6. It is likely attributed to the enhanced energy band gap for the PJ for structure C which sacrifices the carrier tunneling process. Fortunately, it favors the lateral current spreading according to Figure 5(c).
8) It remains unclear why the authors have chosen a current of 180mA and what that really means. If any current value is of importance here, then it should be announced as current density, not as absolute current, as far as the device area is not fixed.
Answer:
(I) The authors thank the referee for the comments. The mesa size of LED is set to 300 ´ 300 μm2. When LED is operated at the current of 180 mA, the responding current density is 200 A/cm2.
(II) There is no significant current crowding effect when the LED is operated at a small current. Therefore, for the purpose of demonstration, a larger current level of 180 mA is selected. Other current levels with even larger value can also be chosen.
9) The manuscript is written in very bad English language. If reconsidered for any further submission, then it must be corrected preferably by a native scientific speaker!
Answer:
The authors thank the referee for the suggestions. English language has been corrected by special scientific speaker according to referees’ suggestions.
10) I have annotated a lot of additional problems as marginal notes in the pdf manuscript for further consideration by the authors.
Answer:
The authors thank the referee for the comments and suggestions. In the revised manuscript, the contents of manuscript have been revised according to the marginal notes in the pdf manuscript.
11) This manuscript describes pure theoretical work without any experimental confirmation. This cannot be easily seen by the reader as it is neither mentioned in the title nor in the abstract, but only at the very end of the introduction.
Answer:
The authors thank the referee for the comments. In order to emphasize the pure theoretical work, the title and abstract of manuscript have been revised according to the referees’ suggestions.
In the revised manuscript, the title and abstract of manuscript are revised as follows.
(I) The title has been revised from “Effect of AlGaN dielectric layer in polarization tunnel junction on performance of AlGaN-based deep ultraviolet light emitting diode” to “Calculation of effect of AlGaN dielectric layer in polarization tunnel junction on performance of AlGaN-based deep ultraviolet light emitting diode”.
(II) Abstract: In this work, AlGaN-based deep ultraviolet (DUV) LEDs with AlGaN as dielectric layers in p+-Al0.55Ga0.45N/AlGaN/n+-Al0.55Ga0.45N polarization tunnel junctions (PTJs) are modelled to promote the carrier tunneling, suppress the current crowding, avoid the optical absorption, and further enhance the performance of LEDs. AlGaN with different Al contents in PTJs are optimized by the APSYS software to investigate the effect of polarization induced electric field (Ep) on the hole tunneling in PTJ. The results indicate that Al0.7Ga0.3N as dielectric layer can realize higher hole concentration and higher radiative recombination rate in MQWs than Al0.4Ga0.6N as dielectric layer. In addition, Al0.7Ga0.3N as dielectric layer has relative high resistance, which can increase lateral current spreading and enhance uniformity of top emitting light of LED. However, the relatively high resistance of Al0.7Ga0.3N as dielectric layer results in increase of forward voltage, so much higher biased voltage is required to enhance hole tunneling efficiency of PTJ. By adopting the PTJ with Al0.7Ga0.3N as dielectric layers, the enhanced internal quantum efficiency (IQE) and output optical power are predicted.

Reviewer 3 Report
Find attached file

Author Response
Response Letter
The authors give special thanks to the referees for their meaningful guidance and constructive suggestions. The manuscript has been revised according to the referee’s comments, and the detailed revision is described in the response to referees. Based on the referees’ comments, the opinions are answered as follows.
(3) Response to Referee 3:
1) Introduction part, the author highly emphasized that the alleviation of current crowding is as important role of TJ and/or PTJ. However, in result part, the current crowding effect is not discussed at all. Does this simulation take such phenomenon into account? Whether take into account or not, I suggest including the discussion on that. I remember that APSYS can incorporate the current crowding effect through further analysis. Please ref”doi.org/10.1364/OE.19.002886”.
Answer:
The authors thank you for your comments and suggestions. In the manuscript, the contents about current crowding have discussed in Figure 5 as follows.
(I) In Figure 5(c): “Fig. 5 (c) shows the comparisons of lateral distributions of hole concentrations and radiative recombination rates along horizontal direction (x axis) and at vertical relative position of the 5th QW (y axis) in LEDs with structures A, B and C. For structure B, the lateral distributions of hole concentrations and radiative recombination rates along horizontal direction (x axis) in LEDs are seriously nonuniform, high at the horizontal positions of 0 ~ 50 μm which is just the bottom of positive electrode, and low at the positions of 50 ~ 200 μm. It means the positions under the positive electrode are the main current flow paths going through the MQWs, which will cause the current crowding and Joule heats. For structures A, B and C, the lateral distribution uniformities of hole concentrations and radiative recombination rates along horizontal direction (x axis) in LEDs are in the following order: structure C > structure A > structure B. The uniformities come from the contributions of high resistances in TJs. structure C with Al0.7Ga0.3N as dielectric layer has higher uniformity than structure B with Al0.4Ga0.6N as dielectric layer, which means relative higher resistance of TJ has a meaningful impact on the current spreading of a LED. ”
(II) In Figure 5(d): “In order to quantitatively compare the lateral distribution uniformities of hole concentrations and radiative recombination rates on the output characteristics of LEDs with structures A, B and C, the integrated intensities of lateral distributions of hole concentrations and radiative recombination rates are characterized. Figure 5 (d) shows the comparisons of integrated intensities of lateral distributions of hole concentrations and radiative recombination rates along horizontal direction (x axis) and at vertical relative position of the 5th QW (y axis) in LEDs with structures A, B and C. The integrated intensities for both hole concentrations and radiative recombination rates are in the following order: Structure C > Structure A > Structure B. Compared with the lateral distributions of hole concentrations and radiative recombination rates along horizontal direction (x axis), the integrated intensities can more accurately reflect the output characteristics of LEDs with structures A, B and C.”
2) When the TJ is employed in LEDs, a current spreading layer (e.g. ITO layer) is unnecessary. Thus, optical absorption differs from conventional device. It could be important when the WPE is obtained. I think that APSYS can incorporate the optical loss in ITO layer through further analysis. If this applied, the difference in results in Figure 7 would be more evident.
Answer:
(I) The authors thank you for your comments and constructive suggestions. In our work, no ITO layer has been used in the APSYS simulation.
The TJ makes a low-resistivity n-type material instead of a high-resistivity p-type material as the top contact with positive electrode of LED, so it is unnecessary to use ITO as a current spreading layer because of its optical absorption.
(II) In the manuscript, the LEE is 0.078, and the output optical power and WPE have been revised accordingly.
3) In page 5, Figure 1 → Figure 2
Answer:
Thank you for your suggestion. In the revised manuscript, Figure 1 in Page 5, line 221 has been revised to Figure 2.

Round 2
Reviewer 2 Report
The reviewer thanks the authors for working on his/her comments and questions. However, still significant problems remain which must be solved before I can agree to publish this manuscript.
Let me say first: After another careful reading, I think I got the main message of the manuscript now: Include a less conductive layer into the p-contact of a LED, then you get better current spreading, hence more light out of an LED which otherwise would be blocked by the electrical contact.
As made clear in more detail below, this is not well described in the manuscript as, for example, the contact geometry, the most critical parameter here, is not really discussed. Moreover, it then remains unclear why we should include a 2nm tunnel "barrier" into such a structure and not just including a slightly less conductive layer into the top 322 nm. Yes, such an Al0.7Ga0.3N layer may partly solve the problem, although the difference to structure A is only minor.
Such a study would not need too complicated discussions of tunneling effects (which finally are not so significant for this problem and on the other hand still fairly unclearly described) but discuss much better things like local current density (completely missing in this manuscript) which then directly leads to locally varying carrier densities.
Let me give some more details:
1) The authors focus very much on current spreading/current crowding in their LED mesa structures. Such phenomena depend very strictly on the lateral shape of the electrical contacts. However, the authors missed to describe the considered contact design. I only can see two black bars in Fig. 5c which might be relevant here. However, these bars or any other details are not explained at all. What do these bars represent? n-type or p-type contact or??? Moreover, the efficiency of a LED (when considering such things as current spreading etc.) depends on the question how much light generated in the quantum wells (e.g. below the non-transparent metal contacts, typically described by IQE) is hindered to exit the LED mesa structure. This may depend on questions like reflectivity of contacts, outcoupling of light to the back-side or to the mesa side walls etc. All these details remain undiscussed in this manuscript.
2) In Fig. 7a, the authors show some data of IQE and tell about some integrated data. "IQE" is the internal quantum efficiency, no matter whether the light exits from the mesa or not. Insofar, it comes before any current spreading is discussed (if not clearly stated differently by the authors). Hence we would expect that the integration goes over the full width of the mesa including the area between the contacts and below the contacts. Then one would expect the same IQE for all three structures A, B, and C. Please clarify this situation.
3) Please also clarify which light you consider when speaking about output power and WPE (Fig. 7). As discussed im my remark (1), this needs a clear description of the contact geometry and even of the light outcoupling paths.
4) On page 9 line 332 you say: "The Al content of AlGaN in TJ determines the resistance". I already tried to get clearer information about this statement in the first round: What exactly leads to an increaded resistance? As far as I understand, it is the lower tunneling probability. It would be good to clarify this.
5) This brings me back to my main concern discussed in the first review report: How did the authors calculate the tunneling probabbility? They now tell that "the shape of tunnel barrier is reflected by electric field intensity in TJ" (whatever an "electric field intensity" is). However, tunneling probability is not only connected with such an electric field (i.e. a potential difference between initial and final state, similarly as argued in the original version by equ. 2, which the authors now have removed without replacement) but mainly depends (as I said before) on the details of the energy barrier shape between initial and final state and the probability of the carrier wave function on either side, connected with the question of occupied states on the initial side and empty states (at same energy) on the final side. It remains unclear whether the authors considered these details. Please also check your description of "degeneracy" on page 7, line 262. The alignment of the bands in Fig. 4 is not a sign or even consequence of degeneracy, but a result of the applied reverse voltage over the tunnel junction. Degeneracy is obtained for heavy doping, visible by the fact that the Fermi level is in the band.
6) On page 5, the authors now have changed the description of the charge density (using the symbol \rho) by just changing the unit from C/cm3 to 1/cm3. Please notice: Charge density must always be indicated in units of C/cm3. What the authors have in mind here are obviously some carrier density or charged state density (i.e. some particle density) or so, typically described with symbols like n or N. Please also check your equation 1 and the symbol \rho in this equation. To get the equation correct, \rho should have the unit cm-2. What is then described by \rho in equ. 1?
7) Unfortunately, the English language is still far from being perfect, making it difficult at some parts to clearly understand what the authors like to say. I have again annotated some problems in the pdf manuscript file, but again: I definitely did not mark all problems.

Author Response
The authors give special thanks to the reviewer for your meaningful guidance and constructive suggestions. The manuscript has been revised according to the reviewer’s comments, and the detailed revision is described in the response. Based on the reviewer’s comments and suggestions, the responses are as follows.
Response:
The reviewer thanks the authors for working on his/her comments and questions. However, still significant problems remain which must be solved before I can agree to publish this manuscript.
Let me say first: After another careful reading, I think I got the main message of the manuscript now: Include a less conductive layer into the p-contact of a LED, then you get better current spreading, hence more light out of an LED which otherwise would be blocked by the electrical contact.
Answer:
The authors thank the reviewer for your comments.
Compared with conventional LED, it makes a low-resistivity n-Al0.55Ga0.45N instead of a high-resistivity p- p+-Al0.55Ga0.45N as the top contact layer with positive electrode of LED, very advantageous to ohmic contact.
As made clear in more detail below, this is not well described in the manuscript as, for example, the contact geometry, the most critical parameter here, is not really discussed.
Answer:
The authors thank the reviewer for your comments and suggestions, and we are sorry for the missed information.
(I) For the contact geometry of LED, the mesa size is set to 300 ´ 300 μm2, with anode size of 300 ´ 50 μm2 and cathode size of 300 ´ 70 μm2. Hence, the following description is added in the revised manuscript as follow.
“The anode of the LED is the ohmic contact on the top layer, and the mesa size was set to 300 ´ 300 μm2, with an anode size of 300 ´ 50 μm2 and a cathode size of 300 ´ 70 μm2.”
(II) In the revised manuscript, Figure 1 has redrawn, in which an anode (top black bar with 300 ´ 50 mm2), a cathode (bottom black bar with 300 ´ 70 mm2), a c-axis, an x-axis, and two rulers in x-axis direction have been added.
Moreover, it then remains unclear why we should include a 2nm tunnel "barrier" into such a structure and not just including a slightly less conductive layer into the top 322 nm. Yes, such an Al0.7Ga0.3N layer may partly solve the problem, although the difference to structure A is only minor.
Answer:
The authors thank the reviewer for your comments. We are sorry that we failed to address our point.
The reviewer is correct that the structure including a less conductive layer into the top 322 nm layers can also improve the current spreading of LED. For example, a PNP-AlGaN current spreading layer has been ever proposed as shown in Figure R1 [27]. However, in this work, we are targeting at the tunnel junctions. Hence, for fair comparison, different tunnel junctions have to be investigated. We then include a 2 nm tunnel “barrier” into such a structure. We take the advantage of compromised effect of the polarization induced electric field and the smaller dielectric constant in the tunnel layer. As a result, both the current spreading effect and the hole injection efficiency can be enhanced.
Such a study would not need too complicated discussions of tunneling effects (which finally are not so significant for this problem and on the other hand still fairly unclearly described) but discuss much better things like local current density (completely missing in this manuscript) which then directly leads to locally varying carrier densities.
Let me give some more details:
1) The authors focus very much on current spreading/current crowding in their LED mesa structures. Such phenomena depend very strictly on the lateral shape of the electrical contacts. However, the authors missed to describe the considered contact design. I only can see two black bars in Fig. 5c which might be relevant here. However, these bars or any other details are not explained at all. What do these bars represent? n-type or p-type contact or??? Moreover, the efficiency of a LED (when considering such things as current spreading etc.) depends on the question how much light generated in the quantum wells (e.g. below the non-transparent metal contacts, typically described by IQE) is hindered to exit the LED mesa structure. This may depend on questions like reflectivity of contacts, outcoupling of light to the back-side or to the mesa side walls etc. All these details remain undiscussed in this manuscript.
Answer:
The authors thank the reviewer for your comments and suggestions, and we are sorry for the missed information.
(I) For the contact geometry of LED, the mesa size is set to 300 ´ 300 μm2, with anode size of 300 ´ 50 μm2 and cathode size of 300 ´ 70 μm2. Hence, the following description is added in the revised manuscript as follow.
“The anode of the LED is the ohmic contact on the top layer, and the mesa size was set to 300 ´ 300 μm2, with an anode size of 300 ´ 50 μm2 and a cathode size of 300 ´ 70 μm2.”
(II) In the revised manuscript, Figure 1 has redrawn, in which an anode (top black bar with 300 ´ 50 mm2), a cathode (bottom black bar with 300 ´ 70 mm2), a c-axis, an x-axis, and two rulers in x-axis direction have been added. In Figure 5c, the background structure of LED with two top black bars has been deleted.
(III) The reviewer is correct that the light propagation is complicated. Our APSYS software calculates the IQE rather than the EQE. Here, in this work, we also conduct FDTD calculations by considering the reflectivity, optical absorption etc., and we get the light extraction efficiency (LEE) of 7.8% for our structures. Once the LEE and IQE are obtained, then the EQE and the optical power are obtained.
2) In Fig. 7a, the authors show some data of IQE and tell about some integrated data. "IQE" is the internal quantum efficiency, no matter whether the light exits from the mesa or not. Insofar, it comes before any current spreading is discussed (if not clearly stated differently by the authors). Hence we would expect that the integration goes over the full width of the mesa including the area between the contacts and below the contacts. Then one would expect the same IQE for all three structures A, B, and C. Please clarify this situation.
Answer:
The authors thank the reviewer for your comments.
As we have mention previously, our APSYS software calculates the IQE rather than the EQE. Here, in this work, we also conduct FDTD calculations by considering the reflectivity, optical absorption etc., and we get the light extraction efficiency (LEE) of 7.8% for our structures. Once the LEE and IQE are obtained, then the EQE and the optical power are obtained. The IQE is indeed obtained by collecting all the photons in the device. However, the photons are generated when electrons are recombined with holes. The radiative recombination process in the active region is decided by the hole concentration in the quantum wells, which is, however, affected by the current flowing paths, i.e., the current spreading effect strongly influences the carrier concentration in the active region. Therefore, the electron-hole recombination process is also affected.
3) Please also clarify which light you consider when speaking about output power and WPE (Fig. 7). As discussed im my remark (1), this needs a clear description of the contact geometry and even of the light outcoupling paths.
Answer:
The authors thank the reviewer for your comments and suggestions.
It is generally accepted that transverse electric (TE)-polarized emission originates from the transition that occurs between the conduction bands and the HH/LH subbands, and that transverse magnetic (TM)-polarized emission originates from the transition between the conduction bands and the CH subbands. The TE polarization has a direction of E⊥[0001], while the TM polarization has a direction of E∥[0001], where E represents the electric field vector of the emitted light.
For the output optical power and WPE, the TE-polarized emission is predominant and the TM-polarized emission is weak.
(I) Considering the optical polarization, when Al content in Al0.4Ga0.6N QW is low, the heavy hole (HH) and light hole (LH) subbands are on the top crystal field (CH) subband in the valence band, TE-polarized emission is then predominant and the TM-polarized emission is weak. The 7.8% LEE is mainly contributed by the TE-polarized light in our FDTD simulations.
(II) Considering the contact geometry of LED, the mesa size is set to 300 ´ 300 μm2, with anode size of 300 ´ 50 μm2 and cathode size of 300 ´ 70 μm2. Therefore, the surface emitting light with TE-polarized emission is predominant.
4) On page 9 line 332 you say: "The Al content of AlGaN in TJ determines the resistance". I already tried to get clearer information about this statement in the first round: What exactly leads to an increaded resistance? As far as I understand, it is the lower tunneling probability. It would be good to clarify this.
Answer:
The authors thank the reviewer for your suggestions.
About the origin of resistance of a TJ, the following descriptions are added and revised in the revised manuscript as follows.
(I) In Figure 5c, “For the comparison of structures A, B and C, the tunneling probability of electrons and Al content of AlGaN in TJs determine the resistance. Structure C has lower tunneling probability of electrons and higher Al content of AlGaN in PTJ than Structure B, which results in higher resistance of PTJ with structure C.”
(II) In Figure 6, the description of “For the comparison of structures A, B and C, the Al content of AlGaN in TJs determines the resistance, and the resistance determines the forward voltage of LED.” has been revised by “For the comparison of structures A, B and C, the tunneling probability of electrons and Al content of AlGaN in TJs determine the resistance, and the resistance determines the forward voltage of LED.”
5) This brings me back to my main concern discussed in the first review report: How did the authors calculate the tunneling probabbility? They now tell that "the shape of tunnel barrier is reflected by electric field intensity in TJ" (whatever an "electric field intensity" is). However, tunneling probability is not only connected with such an electric field (i.e. a potential difference between initial and final state, similarly as argued in the original version by equ. 2, which the authors now have removed without replacement) but mainly depends (as I said before) on the details of the energy barrier shape between initial and final state and the probability of the carrier wave function on either side, connected with the question of occupied states on the initial side and empty states (at same energy) on the final side. It remains unclear whether the authors considered these details. Please also check your description of "degeneracy" on page 7, line 262. The alignment of the bands in Fig. 4 is not a sign or even consequence of degeneracy, but a result of the applied reverse voltage over the tunnel junction. Degeneracy is obtained for heavy doping, visible by the fact that the Fermi level is in the band.
Answer:
The authors thank the reviewer for your comments and suggestions.
(I) The tunneling probability of electrons in a TJ depends on the intensity of Ec, width of TJ, average energy bandgap, and effective tunneling mass. The tunneling probability can be formulated as follows,
where P0 is the tunneling probability, E is the average intensity of Ec, W is the width of TJ, Eg is the average energy bandgap, q is the unit charge, Ñ› is Dirac constant, m* is the effective tunneling mass. For a triangular barrier, E can be drawn in Figure 3, W can be drawn in Figure 4, Eg can be drawn by the relationship of Eg = qEï¹’W, and m* can be drawn by the reduced effective masses of electron and hole for GaN and AlN [56]. The comparisons of width of TJ, average energy bandgap, intensities of Ec, and effective tunneling mass in TJs with structures A, B and C are as follows.
According to Figure 4, the calculated widths of TJs with Structures A, B and C are as follows: |OA| = 8.9 nm, |OB| = 9.5 nm, and |OC| = 8.0 nm, respectively. The widths of TJs with Structures A, B and C are in the following order: Structure B > Structure A > Structure C. According to expressions (2), the short width of the PTJ in Structure C can counteract the large energy barrier, and may help to enhance the tunneling probabilities of the electrons in the PTJ.
According to Figure 4, the calculated energy bandgaps in TJs with structures A, B and C are as follows: Eg = 4.40 eV for Al0.55Ga0.45N, Eg = 4.20 eV for Al0.4Ga0.6N, and Eg = 5.03 eV for Al0.7Ga0.3N, respectively. The average energy bandgaps in TJs with Structures A, B and C are in the following order: Structure C > Structure A > Structure B, as shown in Figure 4. According to expressions (2), the lowest average energy bandgap in PTJ with Structure B can help to enhance tunneling probabilities of electrons in TJ.
According to Figure 3, the intensities of Ec in TJs with structures A, B and C are in the following order: Structure B > Structure C > Structure A in the center peak, and Structure C > Structure A > Structure B at both sides of center. The calculated peak intensities of Ec are as follows: EA = 7.40 ´ 106 V/cm, EB = 9.07 ´ 106 V/cm, and EC = 8.24 ´ 106 V/cm, for Structures A, B, and C in the center peak, respectively. According to expressions (2), the highest average intensity of Ec in PTJ with structure B can help to enhance tunneling probabilities of electrons in TJ.
The effective tunneling mass (m*) has an important impact on the tunneling probabilities of electrons in TJ. m* can be drawn by the reduced effective masses of electron and hole for GaN and AlN [56]. The effective tunneling masses in TJs with Structures A, B and C are in the following order: Structure C > Structure A > Structure B. According to expressions (2), the lowest effective tunneling mass in PTJ with structure B can help to enhance tunneling probabilities of electrons in TJ.
(II) For the physical model of LED with a tunnel junction, the top TJs are operated at a reverse biased voltage when the LEDs are operated at a forward biased voltage. Therefore, the tunnel junction does not have a negative resistance state at the forward voltage. In order to further convince the reviewer, we here will only calculate a p+-GaN/n+-GaN tunnel junction at the forward voltage. The electric field intensity and the energy barrier shape between the initial and the final states can be controlled by the doping concentration in the p+-GaN and n+-GaN layers. Moreover, the doping concentration also affects the number of occupied states and the carrier wave function. Our simulation results show that the tunnel junction has a negative resistance state at the forward voltage. It confirms that the physical model of LED with a tunnel junction do work by considering the parameters mentioned by the reviewer.
(III) The statement of “degeneracy” in the revised manuscript has been removed.
6) On page 5, the authors now have changed the description of the charge density (using the symbol \rho) by just changing the unit from C/cm3 to 1/cm3. Please notice: Charge density must always be indicated in units of C/cm3. What the authors have in mind here are obviously some carrier density or charged state density (i.e. some particle density) or so, typically described with symbols like n or N. Please also check your equation 1 and the symbol \rho in this equation. To get the equation correct, \rho should have the unit cm-2. What is then described by \rho in equ. 1?
Answer:
The authors thank the reviewer for your comments and suggestions. We are so sorry for these errors.
(I) In Equation (1), ρp has been revised by σp in the revised manuscript. According to Equation (1), σp has a unit of 1/cm2.
Moreover, the expression of “ρp is the polarization-induced charge density” has been replaced by “σp is the polarization-induced sheet charge density” in the revised manuscript.
(II) In Figure 2, N is defined as charged state density of space charge, with a unit of 1/cm3, not a unit of C/cm3.
In the revised manuscript, the expressions of “space charge densities” and “polarization charge densities” have been revised by “charged state densities of space charges” and “charged state densities of fixed space charges at interfaces”, respectively.
(III) In Figure 2, more unnecessary contents on the space charges (e.g. ρ5, ρ6, ρ10, ρ11, ρB1, ρB2, ρC1, ρC2 etc) have been deleted in the revised manuscript.
7) Unfortunately, the English language is still far from being perfect, making it difficult at some parts to clearly understand what the authors like to say. I have again annotated some problems in the pdf manuscript file, but again: I definitely did not mark all problems.
Answer:
The authors thank the reviewer for your comments and suggestions.
(I) In the revised manuscript, the contents of manuscript have been revised according to the marginal notes in the pdf manuscript.
(II) Moreover, the revised manuscript has undergone English language editing by MDPI.
Round 3
Reviewer 2 Report
The reviewer thanks the authors for again working on his/her remarks and questions. Unfortunately, some important points are still unresolved. Let me give more details:
1) I asked the authors to describe their findings by using and discussing the local current density. This is still missing. However, this missing description is also important for further points to be clarified, see below.
2) It seems to me that the findings in Fig. 5a and b just reflect the different current densities at this chosen lateral position in the structures A, B, and C. Tunneling does not play a role for explaining these differences besides the fact that different tunnel resistivities lead to different current spreading (which could be easily realized by other means, as discussed in my 2nd review).
3) About my question about the meaning of IQE in your manuscript, you respond "Our APSYS software calculates the IQE rather than the EQE." What still remains unclear is how this is calculated. Let me explain my problem of understanding (again): IQE is defined as ratio between photons generated in the active region and carriers being injected into the active region. In typical cases, uniform carrier injection into a uniform active region is considered, hence IQE is constant over the LED area (for example). In your case, the carrier injection is very non-uniform. Hence also the IQE may be non-uniform. Moreover (and maybe even of more importance): In your LEDs, a lot of photons are generated below the p-contact. As far as I understand your manuscript, these photons do not contribute to the light emitted from the LED (described by, e.g., the WPE), as they get absorbed in the metal contact. This means, that even if the IQE is not depending on the local current density (and hence not of the current spreading), the relation between IQE and EQE depends significantly on the shape of the p-contact and the current distribution (current spreading). If the IQE depends on the local current density, then you must describe it. As you probably know: Typically IQE is small for small current densities (due to nonradiative recombination), finds a maximum at medium current densities and may decrease again at high current densities (due to Auger losses). Hence the situation of IQE (integrally determined for the whole area of the active layers, i.e. below the p-contact and besides the p-contact) may be quite complicated with (certainly) a high current density below the p-contact and a low current density between the contacts. At least, you must announce what you consider as "IQE" in your manuscript: Is it a kind of integrated/averaged value over the region besides the contacts or does it include the p-contact area? Moreover, please let us know whether you consider a constant local IQE with respect to the local current density or not. How do such details lead to differences between the 3 structures A, B, and C?
4) Considering these arguments, the authors must clearly describe why optical output power and IQE are so directly related as shown in Fig. 7a.
5) Answering my question no. 5 from the 2nd review, you now indicate a formula about the tunneling probability. Could you please let me know from which source you got this formula? In this formula you use the parameter "E_g" which typically describes the band gap of a semiconductor. In your explaining text, you later seem to use it in this sence by announcing different values for the 3 different AlGaN layers. However, some lines before you say: E_g=qEW. What is correct? Moreover: How does the band gap of the surrounding Al0.55Ga0.45N layers influence the tunneling probability? This seems not to be considered in this formula.
6) In this explaining text you say "For a triangular barrier, E can be drawn in Figure 3". How can we determine some average electrical field E from Fig. 3? Which values would you extract?
7) Just a remark about language problems: It seems that the English language still can be improved. In my first review I recommended (in the conclusion) to exchange "output optical power" to "optical output power", which you did at this position, but not at many other positions in the text. As I said: I only indicated examples of some problems, not a systematic search of all problems. Even the sentence cited in my question 6 seems to be somewhat unclear language-wise ("E can be drawn in Figure 3"). Unfortunately, I do not have the time to act again as "spell checker" to find other language problems in the current manuscript version.
Author Response
The authors give special thanks to the reviewer for your meaningful guidance and constructive suggestions. The manuscript has been revised according to the reviewer’s comments, and the detailed revision is described in the response. Based on the reviewer’s comments and suggestions, the responses are as follows.
Response:
The reviewer thanks the authors for again working on his/her remarks and questions. Unfortunately, some important points are still unresolved. Let me give more details:
1) I asked the authors to describe their findings by using and discussing the local current density. This is still missing. However, this missing description is also important for further points to be clarified, see below.
Answer:
The authors thank the reviewer for your comments and suggestions.
The current densities along the vertical direction at a relative horizontal position of 100 μm and the lateral distributions of current densities along the horizontal direction (x axis) at a relative vertical position of the fifth QW (c axis) in LEDs for Structures A, B, and C are shown in Figures 1a and 1b (in response letter).
For Figure 1a (in response letter), the current densities along the vertical direction at a relative horizontal position of 100 μm are in the following order: Structure C > Structure A > Structure B, consistent with the distributions of hole concentrations and radiative recombination rates in the MQWs, as shown in Figures 5a and 5b (in manuscript). The high hole concentration in Structure C comes from the enhanced lateral current spreading in the PTJ.
For Figure 1b (in response letter), the lateral distribution uniformities of current densities along the horizontal direction (x axis) at a relative vertical position of the fifth QW (c axis) are in the following order: Structure C > Structure A > Structure B, consistent with the lateral distribution uniformities of hole concentrations and radiative recombination rates in the MQWs, as shown in Figures 5a and 5b (in manuscript).
Structure C has a lower electron tunneling probability and a higher Al content of AlGaN in the PTJ than Structure B, which results in the higher resistance of the PTJ in Structure C. Therefore, Structure C with Al0.7Ga0.3N as the dielectric layer has higher uniformity than Structure B with Al0.4Ga0.6N as the dielectric layer, which means that relatively higher resistance of the TJ has a meaningful impact on the current spreading of an LED.
Figure 1 (a) Current densities along the vertical direction at a relative horizontal position of 100 μm and (b) Lateral distributions of current densities along the horizontal direction (x axis).
2) It seems to me that the findings in Fig. 5a and b just reflect the different current densities at this chosen lateral position in the structures A, B, and C. Tunneling does not play a role for explaining these differences besides the fact that different tunnel resistivities lead to different current spreading (which could be easily realized by other means, as discussed in my 2nd review).
Answer:
The authors thank the reviewer for your comments and suggestions.
(I) The reviewer is correct that the hole concentrations and radiative recombination rates in the MQWs in Figures 5a and b just reflect the current densities at a relative horizontal position of 100 μm in LEDs for Structures A, B, and C. Therefore, the distributions of current densities along the vertical direction at a relative horizontal position of 100 μm are consistent with the distributions of hole concentrations and radiative recombination rates in the MQWs, as shown in Figure 1a (in response letter) and Figures 5a and 5b (in manuscript).
Compared with the lateral distributions of the hole concentrations and radiative recombination rates along the horizontal direction (x axis), the integrated intensities more accurately reflected the output characteristics of LEDs with the different structures, as shown in Figure 5d (in manuscript). The integrated intensities for both the hole concentrations and radiative recombination rates are in the following order: Structure C > Structure A > Structure B.
(II) The uniformities come from the contributions of high resistance in the TJs. For the comparison of structures A, B and C, the tunneling probability of electrons and Al content of AlGaN in TJs determine the resistance. Structure C has lower tunneling probability of electrons and higher Al content of AlGaN in PTJ than Structure B, which results in higher resistance of PTJ and higher lateral distribution uniformity of current spreading.
3) About my question about the meaning of IQE in your manuscript, you respond "Our APSYS software calculates the IQE rather than the EQE." What still remains unclear is how this is calculated. Let me explain my problem of understanding (again): IQE is defined as ratio between photons generated in the active region and carriers being injected into the active region. In typical cases, uniform carrier injection into a uniform active region is considered, hence IQE is constant over the LED area (for example). In your case, the carrier injection is very non-uniform. Hence also the IQE may be non-uniform. Moreover (and maybe even of more importance): In your LEDs, a lot of photons are generated below the p-contact. As far as I understand your manuscript, these photons do not contribute to the light emitted from the LED (described by, e.g., the WPE), as they get absorbed in the metal contact. This means, that even if the IQE is not depending on the local current density (and hence not of the current spreading), the relation between IQE and EQE depends significantly on the shape of the p-contact and the current distribution (current spreading). If the IQE depends on the local current density, then you must describe it. As you probably know: Typically IQE is small for small current densities (due to nonradiative recombination), finds a maximum at medium current densities and may decrease again at high current densities (due to Auger losses). Hence the situation of IQE (integrally determined for the whole area of the active layers, i.e. below the p-contact and besides the p-contact) may be quite complicated with (certainly) a high current density below the p-contact and a low current density between the contacts. At least, you must announce what you consider as "IQE" in your manuscript: Is it a kind of integrated/averaged value over the region besides the contacts or does it include the p-contact area? Moreover, please let us know whether you consider a constant local IQE with respect to the local current density or not. How do such details lead to differences between the 3 structures A, B, and C?
Answer:
The authors thank the reviewer for your comments and suggestions.
(I) Indeed, the APSYS software can only be used to calculate the IQE rather than the EQE. It means that all photons generated in the active region can be extracted from the LED without loss.
In order to obtain the LEE and EQE of LEDs, the FDTD software has been employed by considering the reflectivity, optical absorption etc. Therefore, the LEE of 7.8% can be obtained by the TDTD software for our structures. By adopting the LEE of 7.8%, the optical output power has been calculated by the APSYS software.
(II) For the simulation conducted by the APSYS software, we can think that all photons generated in the active region can be extracted from the LED without loss, and the IQE has nothing to do with the reflectivity, optical absorption from p-contact area. Therefore, the IQE only depends on the ratio between photons generated in the active region and carriers being injected into the active region, and has nothing to do with the reflectivity, optical absorption.
(III) The IQE is an integrated value for our physical model calculated by APSYS software. If we consider a constant local IQE with respect to the local current density, the IQE should be only dependent on the different structures of tunnel junctions.
4) Considering these arguments, the authors must clearly describe why optical output power and IQE are so directly related as shown in Fig. 7a.
Answer:
The authors thank the reviewer for your comments and suggestions.
(I) As the definition, the IQE is defined as a ratio between photons generated in the active region and carriers being injected into the active region, which has nothing to do with the reflectivity, optical absorption from p-contact area. The IQE can be formulated as follows.
Where ηinjection is the injection efficiency, ηradiation is the radiation efficiency, IQW is the current injected into the QW, and Itotal is the total current injected into a LED.
(II) Similarly, the optical output power (Pout) is defined as the work done by light in unit time, which depends on the output photons from an LED and the input electrons into an LED. The optical output power has a relation with the EQE as follows.
Where Pout is optical output power, Iin is the input current, Nphotons is the output photons from an LED, and Nelectrons is the input electrons into an LED.
The optical output power (Pout) has a positive correlation with the EQE. Under the condition of same LEE, the optical output power has a positive correlation with the IQE.
For our physical model calculated by APSYS software, the LEE is set as 7.8%, which has been obtained by the TDTD software. Therefore, the optical output power has a positive correlation with the IQE, as shown in Figure 7a (in manuscript).
5) Answering my question no. 5 from the 2nd review, you now indicate a formula about the tunneling probability. Could you please let me know from which source you got this formula? In this formula you use the parameter "E_g" which typically describes the band gap of a semiconductor. In your explaining text, you later seem to use it in this sence by announcing different values for the 3 different AlGaN layers. However, some lines before you say: E_g=qEW. What is correct? Moreover: How does the band gap of the surrounding Al0.55Ga0.45N layers influence the tunneling probability? This seems not to be considered in this formula.
Answer:
The authors thank the reviewer for your comments and suggestions.
(I) The formula of tunneling probability can be obtained from Reference [J.L Moll. Physics of Semiconductors, chapter 12. McGraw-Hill, 1964].
(II) The tunneling probability can be formulated as follows.
Eg = qEï¹’W
Where P0 is the tunneling probability, E is the average intensity of Ec, W is the width of TJ, and Eg is the average energy bandgap.
According to Figure 4 (in manuscript), the calculated energy bandgaps in TJs are as follows: Eg1 = 4.40 eV for Al0.55Ga0.45N, Eg2 = 4.20 eV for Al0.4Ga0.6N, and Eg3 = 5.03 eV for Al0.7Ga0.3N, respectively.
For Structure A, the energy bandgap in TJ is as Eg1 = 4.40 eV for Al0.55Ga0.45N.
For Structure B, the energy bandgaps in PTJ are as Eg2 = 4.20 eV for Al0.4Ga0.6N and Eg1 = 4.40 eV for Al0.55Ga0.45N.
For Structure C, the energy bandgaps in PTJ are as Eg3 = 5.03 eV for Al0.7Ga0.3N and Eg1 = 4.40 eV for Al0.55Ga0.45N.
The average energy bandgaps (Eg) in TJs with Structures A, B and C are in the following order: Structure C > Structure A > Structure B, as shown in Figure 4 (in manuscript).
In the formula of tunneling probability, the average energy bandgap (Eg) is used, not energy bandgap (Eg1, Eg2, or Eg3). The average energy bandgap (Eg) has a relation with the average intensity (Ec) and the width (W) of TJ.
(III) According to the formula of tunneling probability, the band gaps of surrounding Al0.55Ga0.45N layers in TJs have same impact on the tunneling probability because of the same energy bandgap of Al0.55Ga0.45N.
6) In this explaining text you say "For a triangular barrier, E can be drawn in Figure 3". How can we determine some average electrical field E from Fig. 3? Which values would you extract?
Answer:
The authors thank the reviewer for your comments and suggestions.
(I) According to Figure 3 (in manuscript), the intensities of Ec in TJs with structures A, B and C are in the following order: Structure B > Structure C > Structure A in the center peak, and Structure C > Structure A > Structure B at both sides of center. The calculated peak intensities of Ec are as follows: EA = 7.40 ´ 106 V/cm, EB = 9.07 ´ 106 V/cm, and EC = 8.24 ´ 106 V/cm, for Structures A, B, and C in the center peak, respectively.
(II) According to Figure 3 (in manuscript), the average intensity of Ec can be calculated by the integration at the relative vertical positions between 0.657 and 0.670 μm in Figure 3. The calculated average intensities of Ec are as follows: EA = 4.95 ´ 106 V/cm, EB = 4.60 ´ 106 V/cm, and EC = 5.62 ´ 106 V/cm, for Structures A, B, and C, respectively.
7) Just a remark about language problems: It seems that the English language still can be improved. In my first review I recommended (in the conclusion) to exchange "output optical power" to "optical output power", which you did at this position, but not at many other positions in the text. As I said: I only indicated examples of some problems, not a systematic search of all problems. Even the sentence cited in my question 6 seems to be somewhat unclear language-wise ("E can be drawn in Figure 3"). Unfortunately, I do not have the time to act again as "spell checker" to find other language problems in the current manuscript version.
Answer:
The authors thank the reviewer for your comments and suggestions.
(I) The revised manuscript has undergone English language editing by MDPI.
(II) According to the review’s suggestions, the revised manuscript has been check carefully, and the errors in English language have been revised.
